# Sentinel Lymph Node in Non-Small Cell Lung Cancer: Assessment of Feasibility and Safety by Near-Infrared Fluorescence Imaging and Clinical Consequences

**DOI:** 10.3390/jpm13010090

**Published:** 2022-12-30

**Authors:** Florent Stasiak, Joseph Seitlinger, Arthur Streit, Christophe Wollbrett, Juliette Piccoli, Joelle Siat, Guillaume Gauchotte, Stéphane Renaud

**Affiliations:** 1Department of Thoracic Surgery, Nancy Regional University Hospital, 54500 Nancy, France; 2Department of Pathology and Molecular Biology, Nancy Regional University Hospital, 54500 Nancy, France; 3Research Unit INSERM U1256, NGERE Unit, Lorraine University, 54500 Nancy, France

**Keywords:** lung tumors, occult micrometastases, sentinel lymph node, near-infrared fluorescence, indocyanine green, electromagnetic navigation bronchoscopy

## Abstract

Occult micrometastases can be missed by routine pathological analysis. Mapping of the pulmonary lymphatic system by near-infrared (NIR) fluorescence imaging can identify the first lymph node relay. This sentinel lymph node (SLN) can be analyzed by immunohistochemistry (IHC), which may increase micrometastasis detection and improve staging. This study analyzed the feasibility and safety of identifying SLNs in thoracic surgery by NIR fluorescence imaging in non-small cell lung cancer (NSCLC). This was a prospective, observational, single-center study. Eighty adult patients with suspected localized stage NSCLC (IA1 to IIA) were included between December 2020 and May 2022. All patients received an intraoperative injection of indocyanine green (ICG) directly in the peri tumoural area or by electromagnetic navigational bronchoscopy (ENB). The SLN was then assessed using an infrared fluorescence camera. SLN was identified in 60 patients (75%). Among them, 36 SLNs associated with a primary lung tumor were analyzed by IHC. Four of them were invaded by micrometastases (11.1%). In the case of pN0 SLN, the rest of the lymphadenectomy was cancer free. The identification of SLNs in thoracic surgery by NIR fluorescence imaging seems to be a feasible technique for improving pathological staging.

## 1. Introduction

Non-small cell lung cancer (NSCLC) is one of the most frequent cancers and remains the leading cause of cancer-related deaths in France [1,2]. The cancer prognosis is linked to its stage and therefore to the extension of tumor cells to the pulmonary lymphatic system [3,4]. Surgery has a central place in the management of lung cancer since it represents the cornerstone for localized stages and is part of the therapeutic arsenal for locally advanced stages [5,6]. Lymph node dissection systematically associated with parenchymal resection is one of the recommendations of the main scientific societies [7]. However, lymph node dissection can be related to postoperative morbidity, which is low but may occur due to bleeding, recurrent nerve paralysis, esophageal wounds, chylothorax, aggravation of bronchial congestion by pulmonary denervation, and bronchial fistula by devascularization of the stump [8]. Furthermore, routine pathological analysis techniques using hematoxylin-eosin-saffron (HES) staining can lead to the omission of up to 20 to 30% of micrometastases [9,10,11,12]. The lack of detection of these occult micrometastases can be responsible for an understaging of the disease, and most studies have found that lymph node micrometastases could adversely affect the prognosis of NSCLC patients [13,14].

On the other hand, the introduction of sentinel lymph node (SLN) biopsy has revolutionized the surgical treatment of cutaneous melanoma and breast cancer, becoming a key component in the management of such patients [15,16]. This approach led to a significant reduction in extensive lymphadenectomy-related morbidity [17,18]. The underlying principle is that as the first site of cancer spreads, evaluation of the SLN is expected to be most predictive for wider nodal involvement. SLN can be analyzed more accurately, particularly by immunohistochemistry (IHC), to increase the detection rate of micrometastases [19,20]. Hence, the SLN technique might increase the accuracy of surgical lymph node staging. However, the transposition of this technique in lung cancer is not easy due to the important anatomical variations of the pulmonary lymphatic vessels and the physiological and physical characteristics of the thorax [21]. For over 20 years, thoracic surgeons have worked to apply this technique to lung cancer but with no clinical implications thus far [22,23,24,25,26,27].

However, in the era of the development of new adjuvant treatments and pulmonary segmentectomy, in which one of the conditions for performance is the absence of lymph node metastases [28], the SLN technique may find its place in lung cancer surgery.

More recently, the use of indocyanine green (ICG) and near-infrared (NIR) fluorescence imaging for SLN identification in thoracic surgery has appeared promising [29] and has renewed interest in SLN in lung cancer. The performance of and interest in this approach in the surgical management of localized stage NSCLC are not widely known. Based on these considerations, we hypothesize that SLN detection by NIR fluorescence imaging in thoracic surgery might be a feasible and reliable technique. The main objective of this prospective study was to assess the feasibility of SLN mapping by fluorescence imaging with ICG in lung cancer surgery.

## 2. Materials and Methods

### 2.1. Ethical Statement

This study was approved by our Institutional Review Board. Written consent to participate in the study was obtained from every included patient.

### 2.2. Statistical Analysis

Independent binary proportions or categorized variables were compared via chi-squared testing using Stata software.

### 2.3. Study Design and Outcomes

This was a prospective, observational, single-center trial performed in the thoracic surgery department of the Regional University Hospital of Nancy (France).

The primary endpoint of this study was the identification rate of one or more sentinel lymph node(s) by NIR fluorescence imaging after transpleural or transbronchial injection of ICG.

The secondary endpoints were the upstaging rate and the pathological status of the rest of the lymphadenectomy in cases of negative SLNs. Nodal upstaging was defined as one or more lymph node(s) showing signs of malignancy on anatomopathological examination while they were clinically unscathed on the preoperative assessment.

In addition, adverse effects related to ICG or injection techniques were collected.

An early analysis of the recurrence and survival rates was also performed. Disease-free survival (DFS) was defined as the time elapsed between surgery and radiological and/or pathological proof of local and/or distant recurrence or last follow-up imaging.

### 2.4. Study Population

Adult patients (≥18 years old) with proven or suspected surgically resectable cT1a-cT2b, cN0 (clinical stage I to IIA) NSCLC, no positive lymph nodes on preoperative 18F-fluoro-desoxy-D-glucose positron emission tomography (18F-FDG PET), and written consent were included in this study between December 2020 and May 2022. Each patient’s preoperative thoracic computed tomography (CT) and 18F-FDG PET were reviewed by a radiologist and nuclear radiologist specialized in the field of thoracic oncology. More than 83% of included patients had a peripheral lesion of less than 3 cm and were cN0 on CT and/or PET. According to ESTS recommendations [30], mediastinal staging by EBUS was not performed.

In cases where the lesion did not correspond to these criteria, EBUS was performed but in the majority of cases no biopsy was performed because of the infracentimetric size of the lymph nodes. Pathologic stage was determined according to the eighth edition of the TNM staging system of lung cancer [31].

An operability assessment was systematically carried out as recommended by the ERS/ESTS [32].

The technical aspects of the intervention were assessed, including the surgical approach (open thoracotomy, video-assisted thoracoscopic surgery (VATS) and robot-assisted thoracoscopic surgery (RATS)) and the ICG injection method.

### 2.5. Intraoperative Technique

ICG injection was performed as previously described [29]. In summary, a peritumoral injection of 1 mL of ICG diluted in 20% human albumin to obtain a concentration of 2.5 mg/mL was realized by the transpleural or transbronchial method.

#### 2.5.1. Transpleural Injection

In cases of the direct transpleural approach, ICG was injected through the incision by a 19G fine needle (Arcpoint ^®^, Medtronic, Minneapolis, MN, USA) into the peritumoral area at a depth of at least 1 cm in the parenchyma to limit diffusion of ICG in the chest cavity.

#### 2.5.2. Transbronchial Injection

Navigation in the airways was performed by electromagnetic navigational bronchoscopy (ENB) using the Illumisite ^®^ platform from Medtronic (Minneapolis, MN, USA), as previously described [33]. Once near the lesion, a 19G needle (Arcpoint ^®^, Medtronic, Minneapolis, USA) was inserted through the catheter, and injection of ICG was performed.

In our series, the transbronchial technique was preferentially used. Indeed, transpleural injection was associated with pleural effraction due to needle puncture. This could lead to ICG extravasation in the pleural cavity, making SLN identification more difficult. On the other hand, in the case of a minimally invasive approach, locating small and/or deep tumors to perform a precise injection is sometimes challenging.

The assessment of the SLN by an NIR camera (Visionsens©, Medtronic, Minneapolis, USA) was initiated after at least 5 min of bipulmonary ventilation. Lymphatic mapping by NIR fluorescence imaging was then performed. If an SLN was fluorescent, it was resected, and systematic lymph node dissection was performed in all cases, as recommended [30]. Once resected, the SLN was observed ex vivo with an infrared camera to confirm its avidity with ICG and then sent to the department of pathology apart from the other lymph nodes to perform specific IHC analysis with the anti-cytokeratin antibody AE1/AE3.

## 3. Results

According to our selection criteria, 80 patients were included. The characteristics of the population are disclosed in Table 1.

### 3.1. Primary Outcome

An SLN was identified in 60 of the 80 patients (75%). Of the 20 SLN identification failures, an anatomical cause was found in 14 cases (anthracosis, pachypleuritis, major emphysema), and a technical defect was found in 6 cases (intrapleural injection, fluorescence column malfunction).

A minimally invasive approach (VATS and RATS) was primarily performed (74/80, 92.5%). The injection was performed transpleurally in 22 cases (27.5%) and by ENB in 58 cases (72.5%). Although not statistically significant, there was a better identification rate of the SLN with transbronchial injection compared with transpleural injection (45/58 (77.6%) vs. 15/22 (68.2%)) (*p* = 0.25). Surgical technical aspects are summarized in Table 2.

### 3.2. Secondary Outcomes

The pathological analyses finally revealed 57 NSCLC (42 adenocarcinomas, 15 squamous cell carcinomas), 3 carcinoid tumors, 4 metastatic lesions, and 16 benign lesions.

We hence focused on patients in whom an SLN was identified and analyzed and for whom the pathological analysis concluded a primary lung malignancy. Thirteen patients with a benign lesion and three with metastasis of another cancer were excluded. Of the 44 SLNs associated with a primary lung malignancy, eight were not analyzed (five were intraparenchymal, and three were not analyzed by IHC). Thirty-six patients were thus included (twenty-six adenocarcinomas, seven squamous cell carcinomas, three carcinoid tumors).

There were four cases of SLN invasion by micrometastases detected by IHC, corresponding to 11.1% lymph node upstaging.

In addition, 100% of patients with a negative SLN were pN0 on the final pathological analysis of lymph node dissection. The main results are shown in Figure 1.

In 56 of the 60 cases, the SLN was single (93.3%) and observed in two separated stations in four cases (6.7%). Among single SLNs, five were intraparenchymal (8.9%), 32 were located in N1 hilar and fissure stations (57.1%) and 19 were located in N2 mediastinal stations (33.9%). The exact locations of the SLN are detailed in Table 3 and Table 4.

Finally, we found no adverse events related to the injection of ICG regardless of the injection technique.

### 3.3. Recurrence and Survival

The median follow-up time was 7 months (IQR = 6). Only patients with pN0 primary lung malignancy on definitive anatomopathological analysis and with a CT re-evaluation were included in the survival analysis (*n* = 41).

In the case of pN0 SLNs, disease-free survival (DFS) was better (100% (*n* = 22/22) at 5 and 10 months) than when no SLNs were identified or not analyzed (89.5% (*n* = 17/19) at 5 and 10 months), although the difference did not reach significance (*p* = 0.38). Recurrence occurred in patients with stage IA3 adenocarcinomas. Sites of recurrence were bone and brain in one case and liver, bone, suprarenal, and brain in the other case. More details are available in Figure 2.

## 4. Discussion

The SLN technique remains part of the surgical management of various cancers, such as breast cancer or melanoma, and its transposition to lung cancer has not met a comparable success. Indeed, in recent decades, there have been many attempts to adapt the SLN technique in thoracic surgery using different lymphatic tracers. Little et al. conducted the first trial for thoracic lymphatic mapping in 1999. Using methylene blue as a tracer, they obtained a 47% SLN identification rate (*n* = 17/36). However, blue dye has several limitations, including its wide diffusion to the surface of the lung, its rapid diffusion into the lymphatic network, and difficulties in visualizing the sentinel node among the anthracosic nodes commonly found in the chest [23]. Liptay et al. subsequently used Technecium-99 m as a tracer. With an identification rate of 61.5% of SLNs (*n* = 24/39), they faced different difficulties, such as radiation exposure and poor signal-to-noise ratio at the injection site that disturbed the detection of the lymph nodes [24]. Schmidt et al. and Tiffet et al. tested the combination of an intraoperative injection of methylene blue and Technecium-99 m, with a disparity of the sentinel lymph node identification rate of 81% and 54%, respectively [26,27]. However, with the advent of new technologies such as ENB and new plotters such as ICG, there is renewed interest in this emerging technique in thoracic surgery. Hence, Phillips et al. carried out a trial between 2009 and 2019 aimed at evaluating lymphatic mapping by NIR fluorescence using ICG as a tracer. Compared with methylene blue, the infrared radiation of ICG has the advantages of being invisible to the human eye and of not altering the surgical site. It diffuses less rapidly in the lymphatic capillaries due to dilution with albumin and allows the identification of SLNs even if they are anthracosic. This study identified critical factors important in the success of SLN detection: injection of at least 1 mg of ICG, combination with albumin, and pulmonary ventilation after injection. In patients who provided the optimal combination for injection, SLN was detected in 67.5% of cases (*n* = 27/40) and in 77.4% of cases of a solid nodule (*n* = 24/31) [34]. The results of our study are in line with a 75% identification rate of SLNs. A meta-analysis reported better results than ours with an identification rate between 80 and 84% using fluorescence, but most of the studies analyzed had a low number of patients included (12 to 61) or assessed the SLN ex vivo on a back table after lymphadenectomy [35].

Concerning the two injection techniques, ENB seemed to offer a more physiological diffusion of ICG in lymphatic vessels. We also observed more diffusion of ICG on the lung surface and in the chest cavity with direct injections, probably related to visceral pleural effraction by the needle. These may explain, in addition to the reasons already mentioned, the preferential use of ENB and the better identification rate of the lymph node with this approach in our findings. Nevertheless, the number of patients in these two groups was too small to reach firm conclusions.

The distribution of the SLN location in our study confirms the variable lymphatic drainage of the lung. We also highlighted the presence of SLNs in lymph node areas that are rarely the subject of dissection in daily practice, such as the 3A station. By identifying the most relevant lymph node, the SLN technique could thus improve lymph node staging. The presence of multiple SLNs in our study is not surprising. The existence of SLNs in two different sites has already been described in the literature [36]. This finding can be explained by the interconnections between lymphatic vessels of different lymph nodes, as described by Riquet et al. [21].

In the era of segmentectomy for which a pN0 status is an essential condition for its realization, the identification of the real first lymph node relay seems mandatory. Indeed, due to the great anatomical variability of the pulmonary lymphatic system, the number of skip-N2 metastases in particular can reach 25% to 28.5% in published data [37,38]. It is not uncommon for the first lymph node relay to be directly in the mediastinum. Indeed, Gilmore et al. found 26.9% of skip-N2 for the SLN [29] which is consistent with the 33.9% rate in our study. This highlights the difficulty of predicting the drainage basin of a specific pulmonary zone. It is in this context that the SLN technique finds its place.

With routine pathological analyses by HES, Rusch et al. highlighted that up to 20 to 30% of micrometastases can be missed [12]. Although particularly high, these results were obtained in a very heterogeneous population including stage IA to IIIB NSCLC. In stages IA, Ono et al. found a much lower rate of upstaging of 3.9% with the SLN technique [39]. With a comparable population, our study found a consistent upstaging rate of 11.1% with IHC analyses. Nevertheless, these results confirm the idea that HES staining can underestimate the staging of the disease. However, performing IHC on all lymph nodes retrieved may increase pathological response delay and costs. Interestingly, all the patients with a negative pN0 SLN were cancer free on complementary lymph node dissection, as was previously published [34,40]. This finding supports the idea that pN0 SLNs may be a predictor for the absence of metastatic nodal disease after lymphadenectomy. Hence, SLN could be the only target for IHC if standard HES is negative.

Several animal studies have shown that the primary site of tumor-specific T-cell generation is the lymph node draining the tumor [41,42,43]. A study carried out on murine models with resectable but micrometastatic NSCLC aimed to determine the impact of the resection of lymph nodes draining the tumor (spotted by infrared fluorescence) on the development of pulmonary metastases [44]. It has been shown that the median survival after surgery was statistically better in the group without lymph node resection than in the group with resection of lymph nodes draining the tumor. Thus, lymph node dissection during surgery can potentially affect subsequent disease and response to immunotherapy. These preliminary data question the need for lymph node dissection in the case of pN0 SLN to minimize the risk of immune alteration. However, these results must be balanced because they are not supported by many human studies suggesting no difference or even the opposite. Indeed, studies from the 2000s show a statistically better overall survival and disease-free survival in patients receiving a mediastinal lymph node dissection compared with patients receiving sampling [45]. Other more recent studies found no difference between these two techniques in terms of survival [46].

Our study has some limitations that must be taken into account: the small number of patients, the single institution experience, the learning curve with the ENB technique, which may affect the SLN identification rate, and the short follow-up for DFS. In cases of endobronchial marking, the anesthesia time was increased by 40 min on average compared with a conventional operation, corresponding to the navigation time, then to the change of the intubation tube and the installation of the patient in lateral decubitus. To increase the strength of our study, we are continuing patient inclusion to obtain a larger cohort, and survival analyses will be continued to obtain more robust data. To overcome learning curve bias, we could compare the results over different time periods (e.g., SLN identification rates at the beginning and end of the study). However, while previous studies have studied heterogeneous populations with both localized and advanced stages [34,47], we focused mostly on early stages with small tumors (97.5% stage I), which may better reflect the value of SLNs in real-life practice.

## 5. Conclusions

We have shown that SLN mapping by NIR fluorescence imaging is a safe and feasible technique, leading to a high rate of SLN identification in early-stage lung tumors. This technique may help to improve pathological staging by aiming the SLN for IHC analysis. As lymph node extension is one of the main prognostic factors in NSCLC, the identification of micrometastases and their treatment could improve patient survival. In the future, coupled with rapid on-site examination, it may help to decide whether a segmentectomy can be performed by analyzing the real first lymph node relay. It appears that ENB provides a more physiological diffusion of ICG in the lymphatic drainage area of the tumor compared with the direct injection technique. However, its generalization could be limited by the related costs (i.e., ENB and fluorescence imaging column). Nevertheless, further large prospective studies are needed to confirm these preliminary results, particularly the place of lymphadenectomy in the case of pN0 SLN.

## Figures and Tables

**Figure 1 jpm-13-00090-f001:**
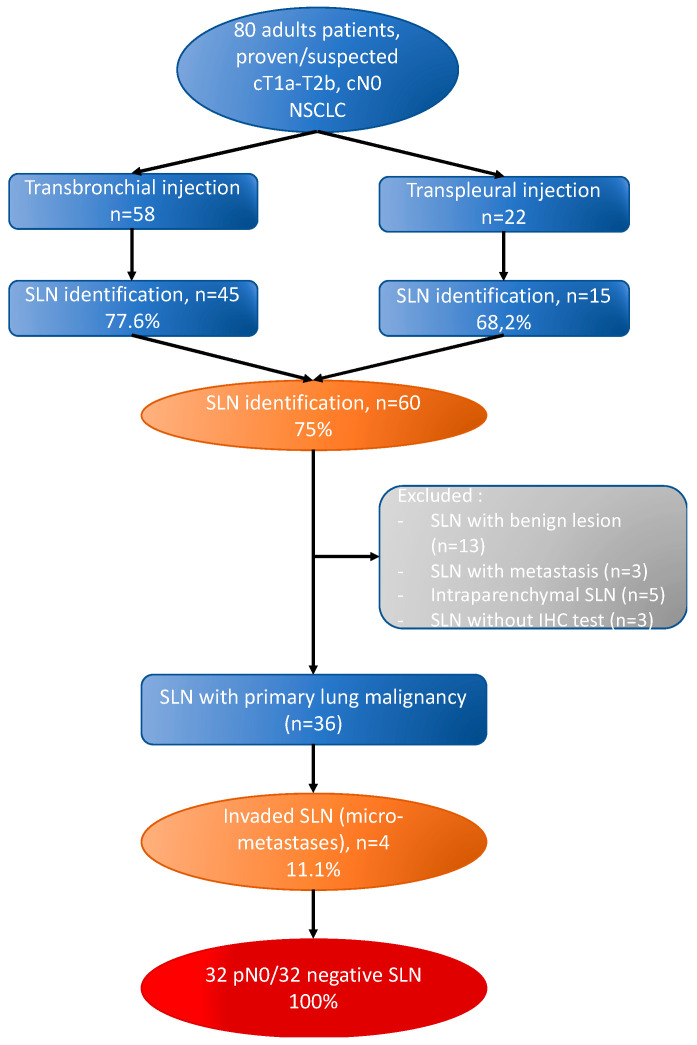
Flowchart of included patients. SLN, Sentinel Lymph Node; IHC, Immunohistochemical.

**Figure 2 jpm-13-00090-f002:**
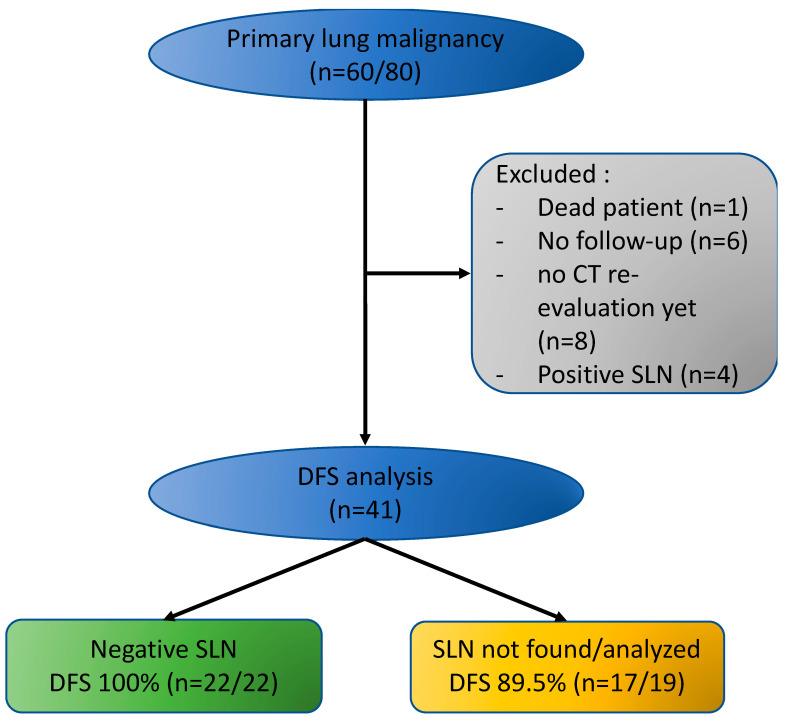
Disease-Free Survival at 5 and 10 months according to SLN pathological status. SLN, Sentinel Lymph Node; DFS, Disease-Free Survival.

**Table 1 jpm-13-00090-t001:** Patients’ characteristics.

Variable	Study Population (*n* = 69)
Age (year), median	68.5 (IQR = 11.25)
Sex, male/female, no. (%)	46/34 (57.5%/42.5%)
BMI (kg/m^2^), median	27.28 (IQR = 6.58)
Charlson score, median	5 (IQR = 2)
WHO score, median	1 (IQR = 0)
FEV1 (% of theoretical), median	86 (IQR = 34)
DLCO (% of theoretical), median	74.5 (IQR = 23.25)
Smoking status, no. (%) - Never - Former - Current	17 (21.25%) 43 (53.75%) 20 (25%)
Tobacco consumption in current or former smokers (pack-years), median	30 (IQR = 15)
Cancer clinical stage, no. (%) - IA1 (cT1aN0) - IA2 (cT1bN0) - IA3 (cT1cN0) - IB (cT2aN0) - IIA (cT2bN0)	18 (22.5%) 37 (46.25%) 12 (15%) 11 (13.75%) 2 (2.5%)
Comorbidities - Coronaropathy, no. (%) - COPD, no. (%) - Diabetes, no. (%) - History of operated lung cancer, no. (%)	15 (18.75%) 24 (30%) 11 (13.75%) 8 (10%)

IQR, Interquartile Range; BMI, Body Mass Index; WHO, World Health Organization; FEV1, Forced Expiratory Volume in one second; DLCO, Diffusing Capacity of the Lung for Carbon Monoxide; COPD, Chronic Obstructive Pulmonary Disease.

**Table 2 jpm-13-00090-t002:** Technical aspects.

Surgical approach (*n* = 69) - Thoracotomy, no. (%) - VATS, no (%) - RATS, no (%)	6 (7.5%) 62 (77.5%) 12 (15%)
Surgical resection (*n* = 69) - Lobectomy, no (%) - Segmentectomy, no (%)	39 (48.75%) 41 (51.25%)
ICG injection (*n* = 69) - Transpleural - Transbronchial	22 (27.5%) 58 (72.5%)
Injection time by ENB * (min), median	10 (IQR = 6)
Identification time of SLN ** (min), median	5 (IQR = 4)

VATS, Video-Assisted Thoracoscopic Surgery; RATS, Robot-Assisted Thoracoscopic Surgery; ICG, Indocyanine Green; min, minutes. * defined by the time between the entry and exit of the bronchoscope in the airway. ** defined by the time between the entry of the camera in the chest cavity and the detection of a fluorescent lymph node.

**Table 3 jpm-13-00090-t003:** Sentinel lymph node locations.

	Single SLN (*n* = 56)
Mediastinal stations, no. (%) - 4 - 5 - 7 - 9	4 (7.1%) 8 (14.3%) 6 (10.7%) 1 (1.8%)
Hilar stations, no. (%) - 10 - 11 - 12	14 (25%) 16 (28.6%) 2 (3.6%)
Intra-parenchymal nodes, no. (%)	5 (8.9%)

SLN, Sentinel Lymph Node.

**Table 4 jpm-13-00090-t004:** Distribution of the different locations of the sentinel lymph node according to the location of the primary tumor.

Location of the Primary Tumor (*n* = 60)	Location of the SLN
RUL (*n* = 21)	Station 4 + 7: 1 Station 4: 3 Station 7: 4 Station 10: 4 Station 11: 4 Station 12: 1 Intra-parenchymal: 4
ML (*n* = 5)	Station 10: 1 Station 11: 3 Station 11 + 12: 1
RLL (*n* = 9)	Station 4: 1 Station 7: 2 Station 7 + 3A: 1 Station 9: 1 Station 11: 4
LUL (*n* = 21)	Station 5: 8 Station 10: 9 Station 11: 3 Intra-parenchymal: 1
LLL (*n* = 4)	Station 7 + 10: 1 Station 11: 2 Station 12: 1
Pathological T stage and location of primary tumor with positive SLN	Case 1: pT1c/LUL/SLN in station 5 Case 2: pT1c/LUL/SLN in station 11L Case 3: pT2a/LUL/SLN in station 5 Case 4: pT2a/RUL/SLN in station 10R

SLN, Sentinel Lymph Node; RUL, Right Upper Lobe; ML, Middle Lobe; RLL, Right Lower Lobe; LUL, Left Upper Lobe; LLL, Left Lower Lobe.

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
