# Peer review of "Sentinel Lymph Node in Non-Small Cell Lung Cancer: Assessment of Feasibility and Safety by Near-Infrared Fluorescence Imaging and Clinical Consequences"

_jpm, 2022, doi:10.3390/jpm13010090_

Round 1

Reviewer 1 Report

This study investigated usefulness of SLN identification by NIR fluorescence imaging. The authors showed that the SLN identification rate was durable, and pN0 by SLN examination was confident. My concerns are as follows.

1.     EBUS guided TBNA was not described in preoperative examination. EBUS guided TBNA was not performed in cN0 patients by chest CT and PET?

2.     Micrometastasis of SLN was observed in 4 cases. Can you provide pathologic T stage, location of the primary tumor and the SLN? This information can be included in Table 4.

Minor

1.     The transbronchial injection section on page 8, the full name for ENB was not given.

Author Response

Response to Reviewer 1 Comments

  1. EBUS guided TBNA was not described in preoperative examination. EBUS guided TBNA was not performed in cN0 patients by chest CT and PET?

Response : More than 83% of included patients had a peripheral lesion, less than 3cm and were cN0 on CT and/or PET. According to ESTS recommendations, mediastinal staging by EBUS was not performed.

In cases where the lesion did not correspond to these criteria, EBUS was performed but in the majority of cases no biopsy was performed because of an infracentimetric size of the lymph nodes.

This information has been added page 7.

  1. Micrometastasis of SLN was observed in 4 cases. Can you provide pathologic T stage, location of the primary tumor and the SLN? This information can be included in Table 4.

Response : this information has been added at the bottom of Table 4.

Minor

  1. The transbronchial injection section on page 8, the full name for ENB was not given.

Response : this has been corrected.

Reviewer 2 Report

We congratulate the authors for conducting this prospective, observational, albeit single-center study, which shows that SLN retrieval by the method used is safe and feasible, leading to a high rate of sentinel node identification in early-stage lung tumors.

The search for the SLN is exciting in thoracic oncology when one knows the success of it in breast or melanoma surgery. However, the complexity of the intrathoracic lymphatic network may pose problems.

If the aim of SLN research is to save lymph node dissection compared to sampling, it should not be forgotten that an abundant literature demonstrates the positive correlation between the number of lymph nodes removed and the vital prognosis, even in early stages of cancer. Complications related to lymph node dissection are quite very rare and are not a reason for quitting this method.

Although the authors define their main objective in the introduction "The main objective was to assess feasibility of SLN mapping", this is not reflected in the outcomes. The primary objective defined is the rate of identification of one or more sentinel nodes. This is not the same thing. Nevertheless, if the priamry objective was the identification rate, we expected to find sensitivity, specificity and accuracy data. Where are they?

Among the secondary objectives, we find the upstaging rate. But there are no results related to this item.

If this article is ever published, we would like to have some more details:

1. Is there a possible relationship between tumor location or size and SLN detection sensitivity?

2. P11: 4 cases of SLN invasion by microetastases detected by IHC have been identified. In which lymph node sites?

3. P19 : the rate of skip-N2 metastases is 25 to 28.5% in the literature. It would be 33.9% in this study. Except that it is not skip metastases but skip-SLN.

4. We would like to know the total time of the anesthesia procedure in this study compared to a conventional operation.

Translated with www.DeepL.com/Translator (free version)

Author Response

Response to Reviewer 2 Comments :

- Although the authors define their main objective in the introduction "The main objective was to assess feasibility of SLN mapping", this is not reflected in the outcomes. The primary objective defined is the rate of identification of one or more sentinel nodes. This is not the same thing. Nevertheless, if the primary objective was the identification rate, we expected to find sensitivity, specificity and accuracy data. Where are they?

Response : You are quite right that the main objective of this study was to assess the feasibility of the sentinel node technique in thoracic oncology and not the rate of sentinel node identification.

By mentioning our sentinel node identification rate, we were implying that this technique was then feasible in thoracic surgery. Especially as this identification rate was rather satisfying (75%), we would not have concluded that the technique was feasible otherwise.

Furthermore, estimating sensitivity and specificity requires a reference technique to compare the explored one. In our case, since sentinel lymph node in thoracic surgery is under investigation, our data cannot be compared to a reference technique. Consequently, specificity and sensitivity cannot be estimated.

- Among the secondary objectives, we find the upstaging rate. But there are no results related to this item.

Response : It is mentioned on page 11 that with 4 sentinel nodes showing signs of malignancy this corresponds to 11% of lymph node upstaging.

- If this article is ever published, we would like to have some more details:

  1. Is there a possible relationship between tumor location or size and SLN detection sensitivity?

Response : This is an excellent question, however the aim of our article was to describe the feasibility of the technique. Your question is currently under investigation, including pathological and molecular data. Nevertheless, regarding the number of analyzes performed, this will be the subject of another article.

  1. P11: 4 cases of SLN invasion by microetastases detected by IHC have been identified. In which lymph node sites?

Response :

Case 1 : pT1c/LUL/SLN in station 5

Case 2 : pT1c/LUL/SLN in station 11L

Case 3 : pT2a/LUL/SLN in station 5

Case 4 : pT2a/RUL/SLN in station 10R

This information has been added at the bottom of Table 4.

  1. P19 : the rate of skip-N2 metastases is 25 to 28.5% in the literature. It would be 33.9% in this study. Except that it is not skip metastases but skip-SLN.

Response : This is a good reflection and a correction has been made on page 18.

  1. We would like to know the total time of the anesthesia procedure in this study compared to a conventional operation.

Response : In cases of transpleural marking, the sentinel lymph node technique has not significantly increase the operation or anesthesia time.

In cases of endobronchial marking, the anesthesia time was increased by 40 minutes on average compared to a conventional operation, corresponding to the navigation time, then to the change of the intubation tube and the installation of the patient in lateral decubitus.

This has been added in the limitations of our study page 19.

Round 2

Reviewer 2 Report

With the answers given to my previous review, this revised document is clear for publication